# Characterization of Stainless Steel Spent Pickling Sludge and Prospects for Its Valorization

Fernando Castro [1],*, Pedro B. Tavares [2], Nuno Cristelo [2], Tiago Teixeira [3], Joana F. Garcia [4] and Nuno M. G. Parreira [4]

1. Metrics Research Centre, University of Minho, Azurém, 4800-058 Guimarães, Portugal
2. Centro de Química-Vila Real, Departamento de Química, ECVA, Universidade de Trás os Montes e Alto Douro, 5000-801 Vila Real, Portugal
3. W2V, Lda, Rua das Alminhas, 900, Calvos, 4810-608 Guimarães, Portugal
4. Bollinghaus Steel, SA, 2431-909 Vieira de Leiria, Portugal
* Correspondence: fcastro@dem.uminho.pt

**Abstract:** Fluorspar is considered a critical raw material for the European Union, due to its industrial uses and lack of sufficient extraction in European countries. It is a source for hydrofluoric acid manufacture, this latter chemical being employed, among other uses, in the pickling of stainless steels. From this latter activity, sludge is generated due to the need for used water treatment. In this article, we report a full characterization of this residue, obtained in an industrial plant in Vieira de Leiria, Portugal. Its chemical and mineralogical characteristics were determined, showing that it is mostly a mixture of calcium fluoride and calcium sulfate with some heavy metals content. Thermal behavior allowed us to determine that the material melts at around 950 °C. The influence of calcining operation on the residue was determined, especially concerned with the leachability of some elements. Taking into account the results of the characterization of this residue, some considerations are presented about the potential for the valorization of this industrial residue.

**Keywords:** stainless steel spent pickling sludge; fluorspar; critical raw material

## 1. Introduction

The European Commission published in 2011 a first list of critical raw materials (CRMs) with the inclusion of 14 commodities that are considered critical for the European Union [1], out of the 41 non-energy, non-agricultural candidate raw materials. The list included antimony, beryllium, cobalt, fluorspar, gallium, germanium, graphite, indium, magnesium, niobium, platinum group metals, rare earth metals, tantalum and tungsten. The list was updated in 2014 to include 20 CRMs out of 54 candidates and again in 2017, to then consider 27 CRMs among 78 candidates. Figure 1 indicates the list of 30 CRMs considered presently by the European Commission.

The purpose of the list is for the European Commission to "flag the supply risks of important materials for the EU economy, ... contributing ... to secure the competitiveness of the EU industrial value chains starting with raw materials in line with the EU industrial policy".

Fluorspar has been included since the first list in 2011. This material is used in metallurgy, namely in aluminum extraction and in steelmaking as a desulfurization agent. Other uses include gasoline, insulating foams, refrigerants, uranium fuel, steel castings, welding rod coatings, glass manufacture, cement and enamel production [3]. It is the source for the manufacturing of hydrofluoric acid. China is currently the first producer of fluorspar with about 65% of the World's production, followed by Mexico and Mongolia with around 10% each.

| 2020 Critical Raw Materials (30) | | | |
|---|---|---|---|
| Antimony | Fluorspar | Magnesium | Silicon Metal |
| Baryte | Gallium | Natural Graphite | Tantalum |
| Bauxite | Germanium | Natural Rubber | Titanium |
| Beryllium | Hafnium | Niobium | Vanadium |
| Bismuth | HREEs | PGMs | Tungsten |
| Borates | Indium | Phosphate rock | Strontium |
| Cobalt | Lithium | Phosphorus | |
| Coking Coal | LREEs | Scandium | |

**Figure 1.** The 2020 List of Critical Raw Material [2].

European production is limited to Spain and Germany with less than 3% of the World's needs. Mexico is the country with the most significant known reserves [3]. The European Union's biggest importers are Italy, Belgium, Germany, Finland, France, Spain, Sweden, Greece and Poland, and the whole imports of the EU represent around 10% of the World's total production [4]. Fluorspar is hence an important raw material for the European industry and its dependence on imports makes it a critical raw material. Hydrofluoric acid, by its side, is also a critical chemical, used for the production of fluorocarbons, as a catalyst in petroleum alkylation, for the manufacture of inorganic fluorine compounds and in metal treatment, such as pickling of stainless steel.

The treatment of stainless steel rods to prepare their surface for hot rolling involves pickling with hydrofluoric/sulfuric acid solutions. The acidic solutions are recirculated till the acidity and the amount of iron rends it difficult for further pickling [5]. Then, the wastewater is treated. In the plant of Bollinghaus Steel, SA, at Vieira de Leiria, Portugal, the treatment of the resulting wastewater is carried out by adding a slaked lime slurry to neutralize the acidity. By this procedure, sludge is precipitated and then dewatered in filter press units. The cake is then a final residue of the process, which is currently an object of disposal in landfills.

The neutralization process is performed till a pH value between 8.0 and 8.5 is reached. The involved reactions are:

$$2\,HF + Ca(OH)_2 \;\rightarrow\; CaF_2 + 2\,H_2O$$

and

$$H_2SO_4 + Ca(OH)_2 \;\rightarrow\; CaSO_4 + 2\,H_2O$$

In both cases, the compounds calcium fluoride and calcium sulfate are precipitated.

By means of the neutralization of wastewater, the solubility of metal hydroxides is reduced [6–11] as shown in Figure 2.

As a consequence, iron and other heavy metal hydroxides are expected to precipitate along with calcium fluoride and calcium sulfate.

The so-obtained sludge is considered a waste, classified by the European Commission [12] under code LER 11 01 09 or 11 01 10. The first of these is considered hazardous waste because it contains hazardous substances exceeding regulatory limits. The second code applies when there are no hazardous substances in the chemical composition of the sludge. Even when the sludge is considered non-hazardous, the environmental impact of its disposal is not negligible, even after a stabilization procedure [13]. In view of these environmental aspects and because the metal and the calcium compound contents may be interesting, the potential to valorize this type of sludge is being studied by several researchers in the following directions:

- Recovery of valuable metals by means of hydrometallurgical routes [14–16];
- Recovery of metals by pyrometallurgical and smelting processes [17–20];
- Use in the manufacture of construction materials [21–24].

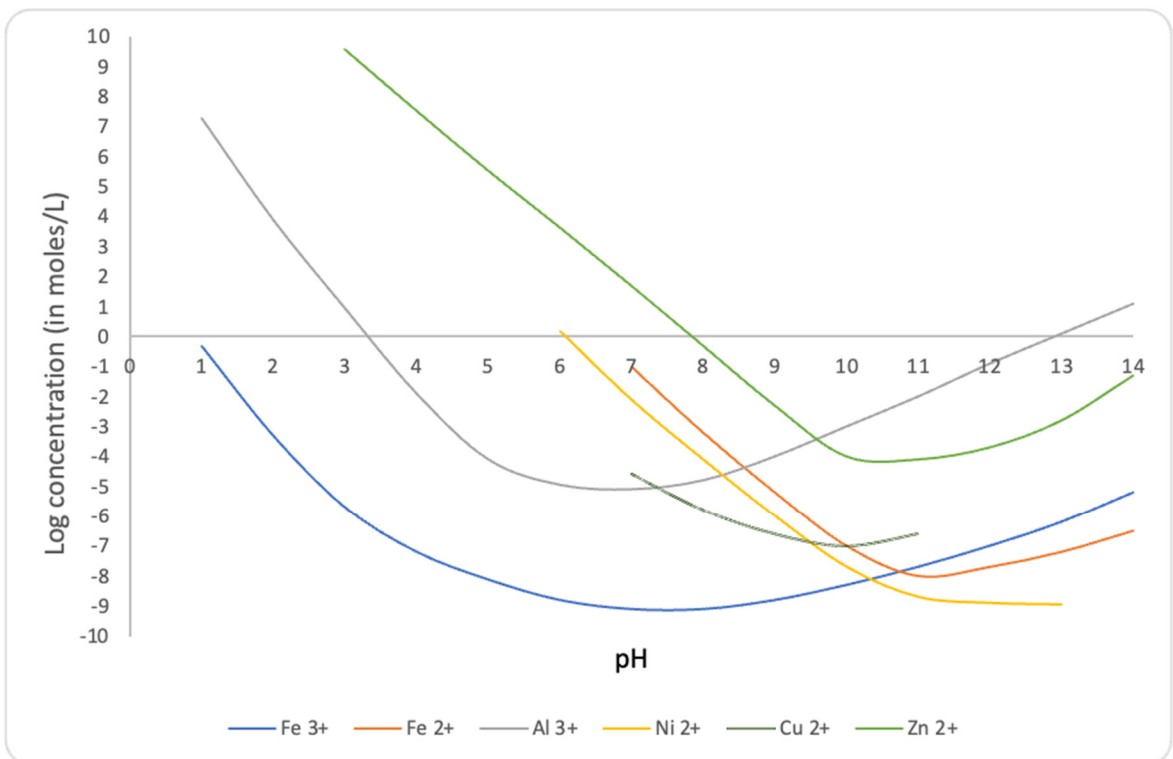

**Figure 2.** Solubilities of some metal hydroxides as a function of pH. Our calculations based on data from [6–11].

Due to the low content in valuable metals, such as Nickel and Chromium, and the high sulfur content of the sludges, its use as a secondary raw material for metal extraction is still doubtful economics. However, due to the increasing prices of metals, an economic interest in these procedures could increase. The use in the manufacture of construction materials is often a low-grade valorization scheme, as the main purpose is to replace other natural raw materials, avoiding disposal in landfills. The recovery of fluor compounds may be a promising route, as fluorspar is considered a critical raw material in the EU.

## 2. Materials and Methods

With the purpose of evaluating the recycling potential for this wastewater treatment residue, we performed a characterization under different techniques.

To have a representative source of produced sludges, samples of around 1 kg were collected at the wastewater treatment facilities of Bollinghaus Steel, S.A. manufacturing plant, in Vieira de Leiria, Portugal, on 5 different days. The samples were collected randomly at the exit of the filter press. The characterization involved the following methods:

- Determination of the moisture content;
- Determination of the loss on ignition at 900 and 1000 °C;
- Chemical analysis determination;
- X-ray diffraction;
- Observation with scanning electron microscope;
- Simultaneous Differential Thermal Analysis and Thermogravimetry (DTA/TGA);
- Infrared spectroscopy;
- Water leaching and analysis of eluate;
- Ammonia citrate leaching test.

Sub-samples for above-indicated characterizations were obtained from the samples collected in the plant by dividing them into 4 parts. Then, 200 g were taken from each part and the 4 were mixed again by means of putting the whole in a rotary jar (30 rotations per

minute) for 1 h to homogenize the material. The necessary material for each characterization was collected randomly from the as-prepared sub-samples.

For each type of characterization, the following methods were employed.

### 2.1. Determination of the Moisture Content

The moisture was determined by drying a 5 g sub-sample of each material at $105 \pm 2$ °C for at least 12 h until the mass of the sample remained constant. The weight was measured in a 0.1 mg precision balance.

### 2.2. Determination of the Loss on Ignition at 900 and 1000 °C

The loss on ignition was determined in an oven, for 1 h, at the indicated temperature $\pm 10$ °C. The sub-sample size was 10 g and the weight was measured in a 0.1 mg precision balance.

### 2.3. Chemical Analysis Determination

The chemical analysis was performed on previously dried samples, obtained according to the procedure indicated above for the determination of moisture content. X-ray fluorescence spectrometry analysis was obtained in Hitachi EA1000VX equipment (Hitachi Ltd., Tokyo, Japan). Carbon and sulfur were determined by elemental analysis in a C-S 200 spectrometer (LECO Corporation, St. Joseph, MI, USA).

### 2.4. X-ray Diffraction

X-ray diffraction was performed on samples previously dried at 60 °C for 6 h and also on samples calcined at 900 °C according to the procedure described for the loss on ignition determination. X-ray diffraction (XRD) patterns were recorded at room temperature using a Panalytical MPD diffractometer equipped with an X´Celerator detector and secondary monochromator (Malvern Panalytical, Almelo, The Netherlands) in Bragg–Brentano geometry. CuK$\alpha$1.2 radiation, $\lambda$1 = 1.5406 Å, $\lambda$2 = 1.5444 Å, a step size of 0.017° and 100 s/step were used. Phases quantification was calculated using Rietveld refinements with HighScore$^{\circledR}$ 4.8 software and COD database (Malvern Panalytical, Almelo, The Netherlands).

### 2.5. Observation in Scanning Electron Microscope

The observation in scanning electron microscopy was performed on samples dried at 105 °C and also on calcined samples. Scanning Electron Microscopy (SEM) was performed using a JEOL, series 6010LV with EDS INCAx-Act, PentaFET precision, (Oxford Instruments, Abingdon, UK).

### 2.6. Simultaneous Differential Thermal Analysis and Thermogravimetry (DTA/TGA)

A SDT 2960 Simultaneous DSC-TGA (TA Instruments, New Castle, DE, USA) was employed for these determinations. The atmosphere was argon and the heating rate was 10 °C per minute, till 1 100 °C was reached. Then, the material was kept at this temperature for 30 min and then cooled at the same rate as for heating. The samples were used in their initial condition, not previously dried.

### 2.7. Infrared Spectroscopy

Fourier-transform infrared spectroscopy was carried out on a Bruker Alpha Platinum-ATR spectroscope (Bruker Optics GmbH & Co. KG, Ettlingen, Germany). Samples were previously dried or calcined, according to the above-mentioned procedures.

### 2.8. Water Leaching and Analysis of Eluate

The water leaching followed the prescribed in the UNE-EN 12457-4 standard. This implied the leaching in distilled water in a solid/liquid ratio = 1/10. During 24 h in a rotary device at a speed of 7.5 rotations per minute. Then, the solids were filtered under vacuum

with 0.45 μm PTFE membranes and the eluate evaporated to determine the total solids, their composition being obtained by X-ray fluorescence spectrometry.

### 2.9. Ammonia Citrate Leaching Test

The procedure for the ammonia citrate leaching evaluation followed a similar procedure as the one considered for determination of the extraction of phosphorus from fertilizers [25]. It considered the neutral ammonium citrate solution at a concentration of 234 g/L, a 1/100 solid/liquid ratio, leaching at 15 °C for 1 h. Agitation was performed using a magnetic bar at moderate rotation speed. A sub-sample of 10 g was employed. Concentration of the elements was obtained by ICP spectrometry in the laboratory of ALS Czech Republic, s.r.o., in Ceska Lipa

## 3. Results and Discussion

### 3.1. Determination of the Moisture Content and Loss on Ignition

Table 1 resumes the results for the determination of moisture content and of loss on ignition (L.o.I).

**Table 1.** Moisture content and loss on ignition of the samples collected on 5 consecutive days.

| Sample | Moisture (%) | L.o.I. 900 °C (%) | L.o.I. 1000 °C (%) |
|---|---|---|---|
| 14.02.2022 | 37.9 | 51.6 | 55.5 |
| 15.02.2022 | 48.1 | 55.3 | 57.9 |
| 16.02.2022 | 44.0 | 57.9 | 58.1 |
| 17.02.2022 | 40.5 | 62.0 | 60.7 |
| 18.02.2022 | 43.4 | 52.6 | 52.2 |
| Average | 42.8 | 55.9 | 56.9 |
| Standard Deviation | 3.8 | 4.2 | 3.2 |

These results indicate that the major part of the weight loss during heating is due to moisture evaporation. Loss on ignition at 900 and 1000 °C indicates similar results. Loss on ignition at 900 °C is mainly due to the calcination of gypsum. An additional mass loss is also expected to occur around 120–150 °C due to the evaporation of the combined water of gypsum.

### 3.2. Chemical Analysis

Table 2 presents the chemical analysis of the dried samples, with metals and sulfur indicated in the oxidic form.

**Table 2.** Chemical analysis of the previously dried samples obtained by XRF spectrometry.

| Component | 14.02.2022 | 15.02.2022 | 16.02.2022 | 17.02.2022 | 18.02.2022 | Average | Stand. Dev. |
|---|---|---|---|---|---|---|---|
| CaO | 34.8 | 35.2 | 35.3 | 34.7 | 34.8 | 35.0 | 0.27 |
| $Fe_2O_3$ | 17.8 | 17.0 | 16.8 | 17.1 | 17.4 | 17.2 | 0.39 |
| $SO_3$ | 21.0 | 21.4 | 20.7 | 20.7 | 20.7 | 20.9 | 0.31 |
| F | 17.6 | 16.4 | 16.9 | 17.2 | 17.2 | 17.1 | 0.45 |
| Cl | 0.20 | 0.21 | 0.23 | 0.21 | 0.19 | 0.21 | 0.01 |
| $SiO_2$ | 0.24 | 0.24 | 0.25 | 0.24 | 0.24 | 0.24 | 0.004 |
| C | 0.25 | 0.22 | 0.25 | 0.26 | 0.30 | 0.26 | 0.03 |
| $Al_2O_3$ | 0.43 | 0.75 | 1.11 | 0.85 | 0.28 | 0.68 | 0.33 |
| MgO | 0.56 | 1.02 | 0.85 | 0.88 | 0.72 | 0.81 | 0.17 |

**Table 2.** *Cont.*

| Component | 14.02.2022 | 15.02.2022 | 16.02.2022 | 17.02.2022 | 18.02.2022 | Average | Stand. Dev. |
|---|---|---|---|---|---|---|---|
| $Na_2O$ | 0.30 | 0.36 | 0.27 | 0.30 | 0.27 | 0.30 | 0.04 |
| $K_2O$ | 0.22 | 0.22 | 0.23 | 0.24 | 0.22 | 0.23 | 0.01 |
| $TiO_2$ | 0.12 | 0.12 | 0.12 | 0.11 | 0.12 | 0.12 | 0.004 |
| $MnO$ | 0.22 | 0.21 | 0.21 | 0.20 | 0.22 | 0.21 | 0.01 |
| $Cr_2O_3$ | 2.97 | 3.64 | 3.70 | 3.80 | 3.93 | 3.61 | 0.37 |
| $NiO$ | 2.44 | 2.30 | 2.23 | 2.33 | 2.44 | 2.35 | 0.09 |
| $CuO$ | 0.66 | 0.56 | 0.62 | 0.72 | 0.86 | 0.68 | 0.11 |
| $CoO$ | 0.09 | 0.07 | 0.09 | 0.08 | 0.08 | 0.08 | 0.01 |
| $MoO_2$ | 0.09 | 0.09 | 0.09 | 0.09 | 0.09 | 0.09 | 0 |

These results indicate a fairly good homogeneity of composition. The major elements are calcium, fluorine, sulfur and iron. It is worth mentioning the presence of more than 2% of nickel oxide, which could be relevant for the purpose of valuable metal recovery.

### 3.3. X-ray Diffraction

Figure 3 presents the diffractograms of the five samples, dried at 60 °C.

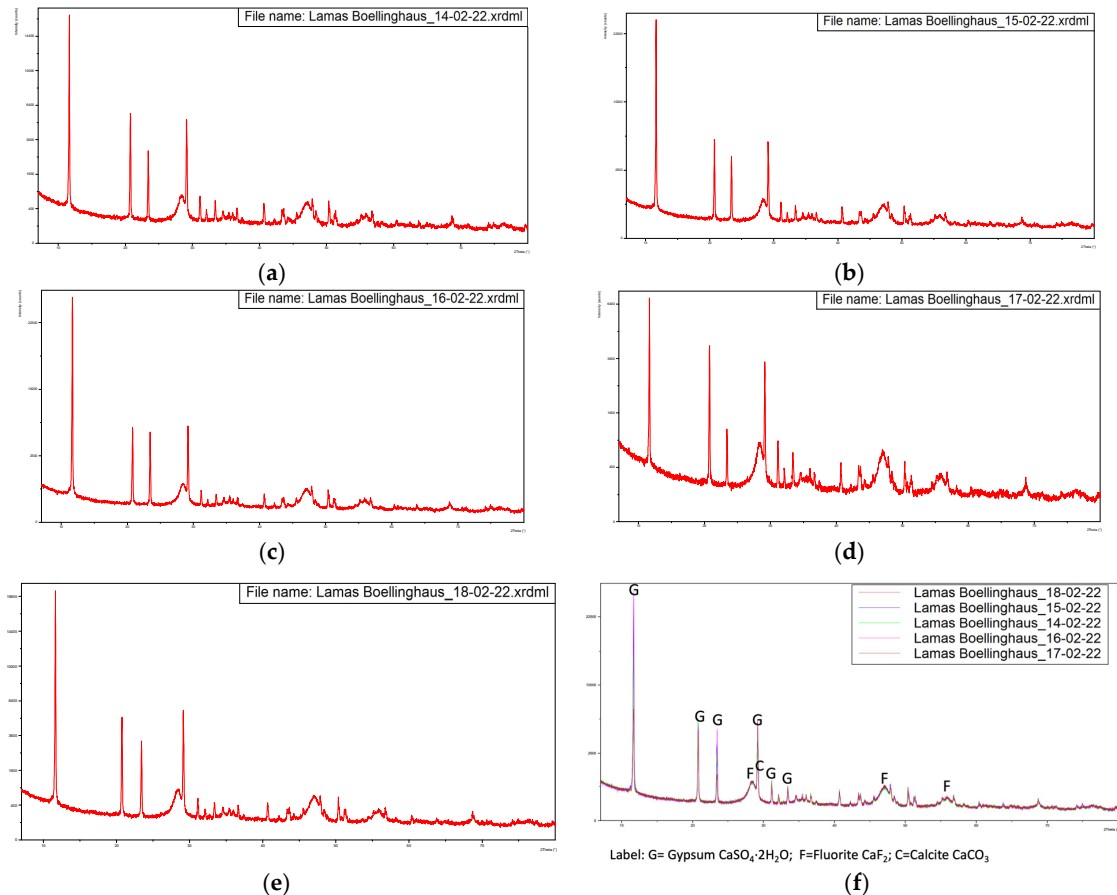

**Figure 3.** Diffractograms of the five samples dried at 60 °C with overlapping of the same. (**a**) Sample from 14-02-2022, (**b**) Sample from 15-02-2022, (**c**) Sample from 16-02-2022, (**d**) Sample from 17-02-2022, (**e**) Sample from 18-02-2022, (**f**) Overlapping of the previous diffractograms with identification of the main peaks.

These diffractograms suggest a very similar mineral structure for all samples.

The Rietveld refinement of the diffractograms allowed us to estimate the compound compositions indicated in Table 3.

**Table 3.** Estimated compound compositions of the samples dried at 60 °C obtained by the Rietveld refinement method.

| Compound | 14.02.2022 | 15.02.2022 | 16.02.2022 | 17.02.2022 | 18.02.2022 |
|---|---|---|---|---|---|
| $CaSO_4 \cdot 2H_2O$ | 57.8 | 54.3 | 49.8 | 49.6 | 49.2 |
| Fluorite ($CaF_2$) | 31.5 | 34.7 | 35.9 | 35.1 | 37.4 |
| Magnetite ($Fe_3O_4$) | 0.5 | 0.5 | 0.4 | 0.3 | 0.5 |
| Calcite ($CaCO_3$) | 9 | 9,4 | 13.1 | 14.4 | 11.8 |
| Guyanaite (CrOOH) | 0.2 | 0.2 | 0.2 | 0.2 | 0.2 |
| $Cr_2O_3$ | 0.4 | 0.4 | 0.3 | n,d | 0.4 |
| $MgF_2$ | 0.5 | 0.5 | 0.4 | 0.4 | 0.5 |

This indicates the major presence of calcium sulfate di-hydrate, fluorite and calcite. The fact that no relevant peaks were identified for the iron-bearing phases suggests that this element is present in the sludge in amorphous form.

For samples calcined at 900 °C, Figure 4 presents the obtained diffractograms.

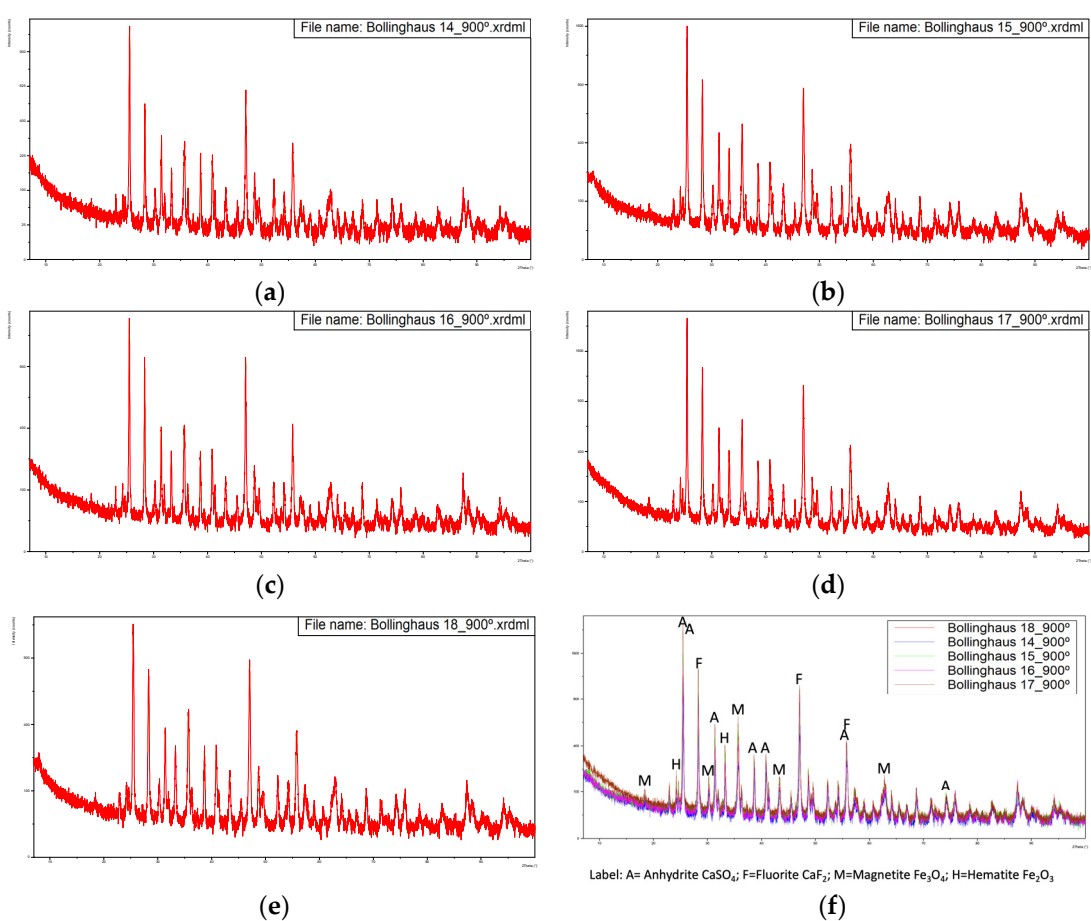

**Figure 4.** Diffractograms of the five samples calcined at 900 °C with overlapping of the same. (**a**) Sample from 14-02-2022, (**b**) Sample from 15-02-2022, (**c**) Sample from 16-02-2022, (**d**) Sample from 17-02-2022, (**e**) Sample from 18-02-2022, (**f**) Overlapping of the previous diffractograms with identification of the main peaks.

The Rietveld refinement of the diffractograms allowed us to estimate the compound compositions indicated in Table 4.

**Table 4.** Estimated compound compositions of the samples calcined at 900 °C obtained by the Rietveld refinement method.

| Compound | 14.02.2022 | 15.02.2022 | 16.02.2022 | 17.02.2022 | 18.02.2022 |
|---|---|---|---|---|---|
| Anhydrite ($CaSO_4$) | 53.4 | 48.2 | 47.7 | 46.6 | 45.8 |
| Fluorite | 24.9 | 28.5 | 30.7 | 30.7 | 30.5 |
| Magnetite | 11.5 | 11 | 10.4 | 10.4 | 11.5 |
| Hematite | 9.1 | 11.2 | 11.4 | 11.4 | 11.3 |
| $NiCrF_6$ | 0.8 | 0.7 | 0.8 | 0.8 | 0.7 |
| Srebrodolskite ($Fe_2Ca_2O_5$) | 0.3 | 0.3 | 0.2 | 0.2 | 0.2 |

The major compounds are anhydrite, fluorite and iron oxides, magnetite and hematite.

### 3.4. Observation in a Scanning Electron Microscope

Figure 5 indicates a view of the sample dried at 105 °C.

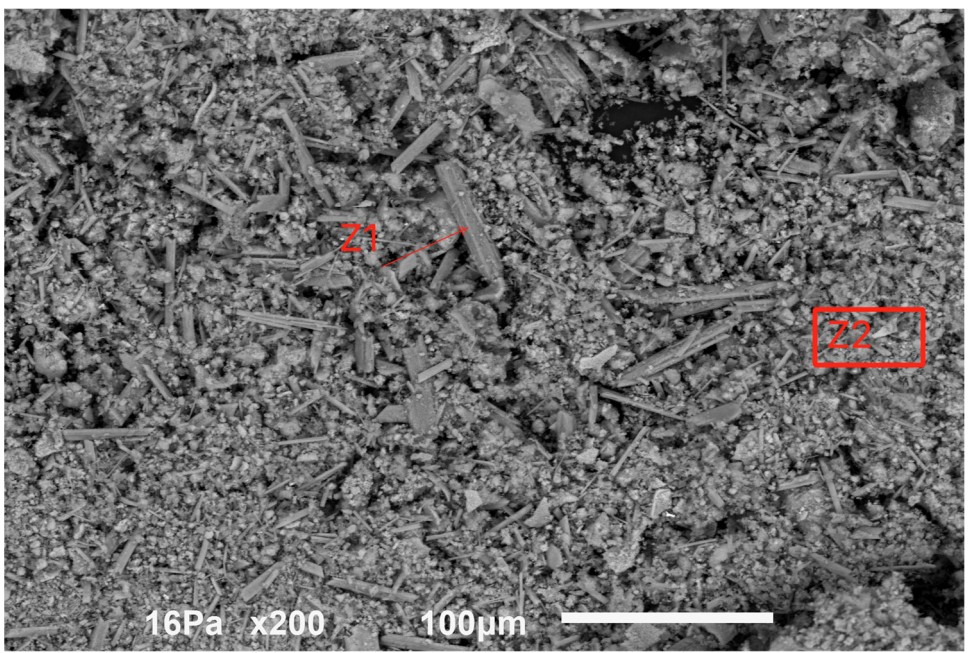

**Figure 5.** General aspect of the sample dried at 105 °C.

The material is composed of a very fine round-shaped powder of less than 10 μm, most of the particles being in the range of 1–5 μm, with some tubular-like crystals of higher dimensions reaching sometimes more than 50 μm. The energy dispersive spectrometry analysis (EDS) indicated the spectra for the tubular-like crystals (Z1 in the above figure) and for the round-shaped grains (Z2), as is shown in Figure 6.

These chemical analyses indicate that the tubular-like crystals are almost pure calcium sulfate, while the finer round-shaped grains are a mixture of calcium sulfate and calcium fluoride with the other metals, Fe, Cr and Ni.

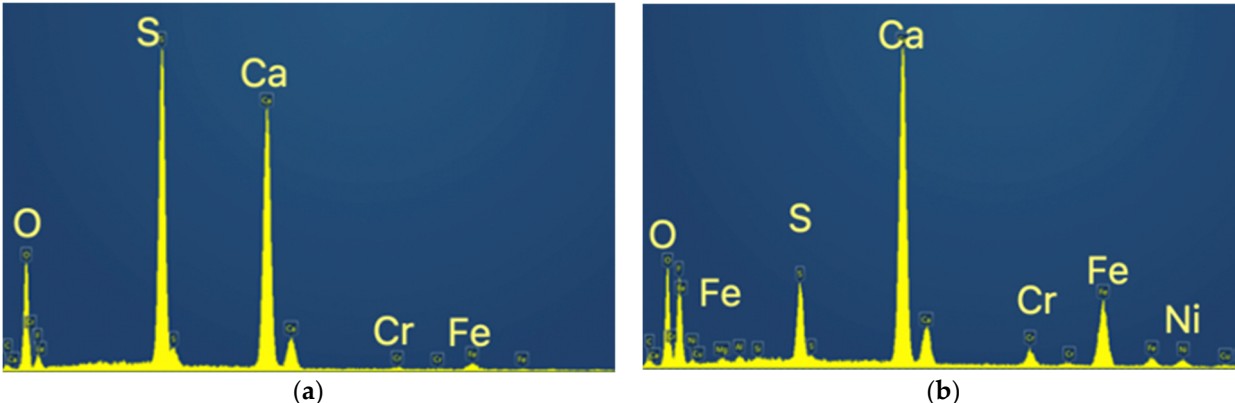

**Figure 6.** EDS spectra for the tubular-like crystals and for the round-shaped grains. (**a**) Spectrum of Z1 (tubular crystals); (**b**) Spectrum of Z2 (round shaped grains).

For the sample calcined at 1000 °C, Figure 7 indicates a compact structure with some internal porosity. Three major phases were identified, indicated as Z1 (the matrix), Z2 and Z3, whose EDS spectra are indicated in Figure 8.

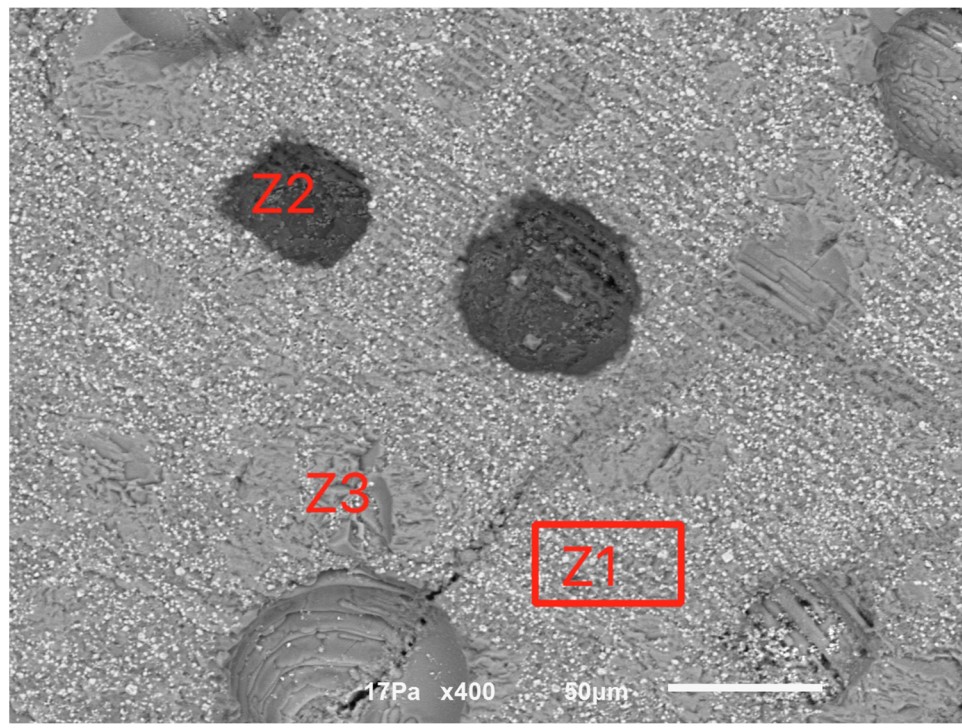

**Figure 7.** General view of the sample calcined at 1000 °C.

The matrix concentrates the heavier metals, such as Fe, Cr and Ni, while Z2 is a calcium sulfate and fluoride mixture and Z3 seems to be mainly calcium fluoride.

### 3.5. Simultaneous Differential Thermal Analysis and Thermogravimetry (DTA/TGA)

Figure 9 presents the graph of DTA/TGA determination of the sample collected on 18.02.2022.

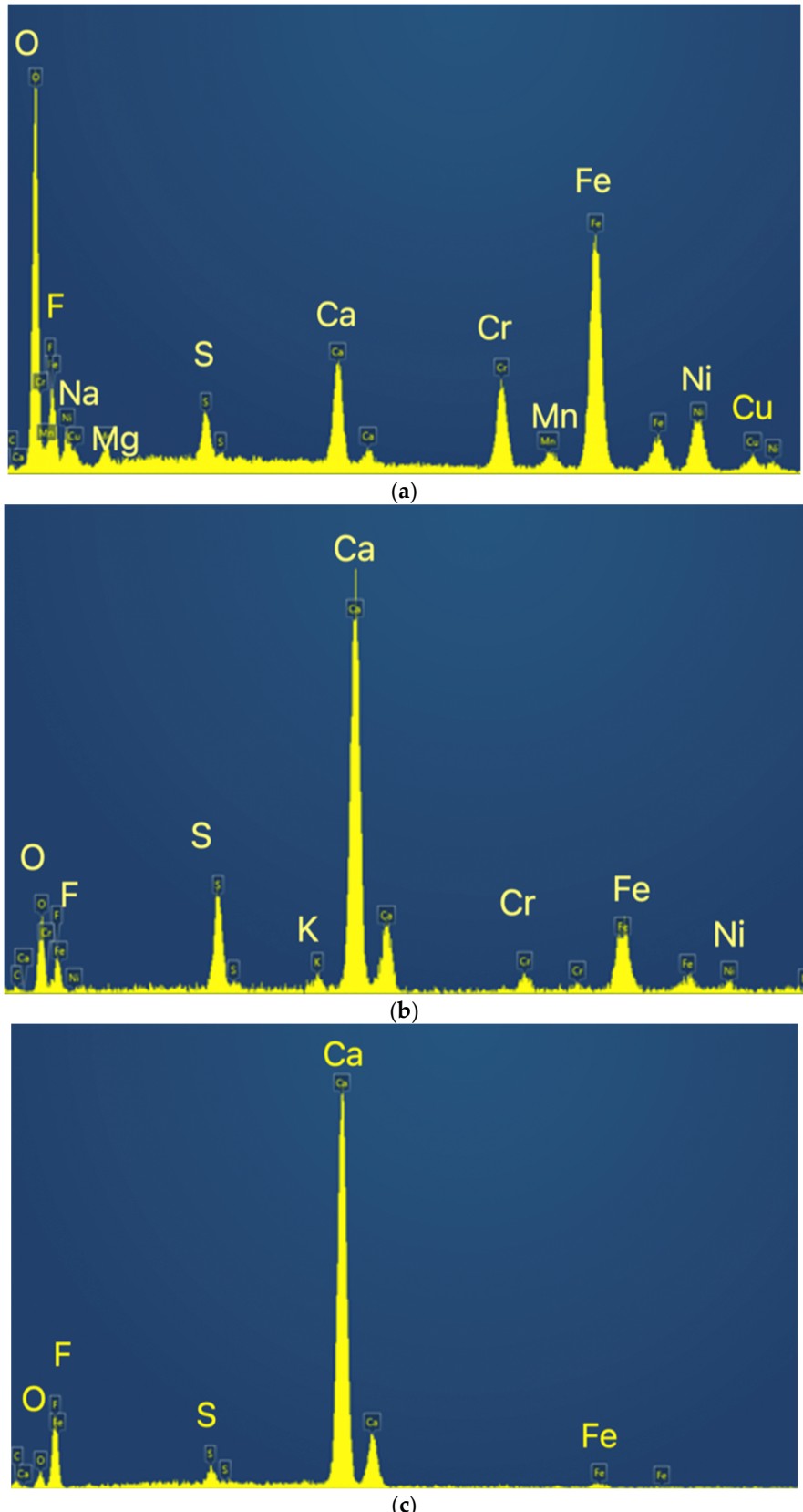

**Figure 8.** EDS spectra for the identified phases. (**a**) Spectrum of Z1; (**b**) Spectrum of Z2; (**c**) Spectrum of Z3.

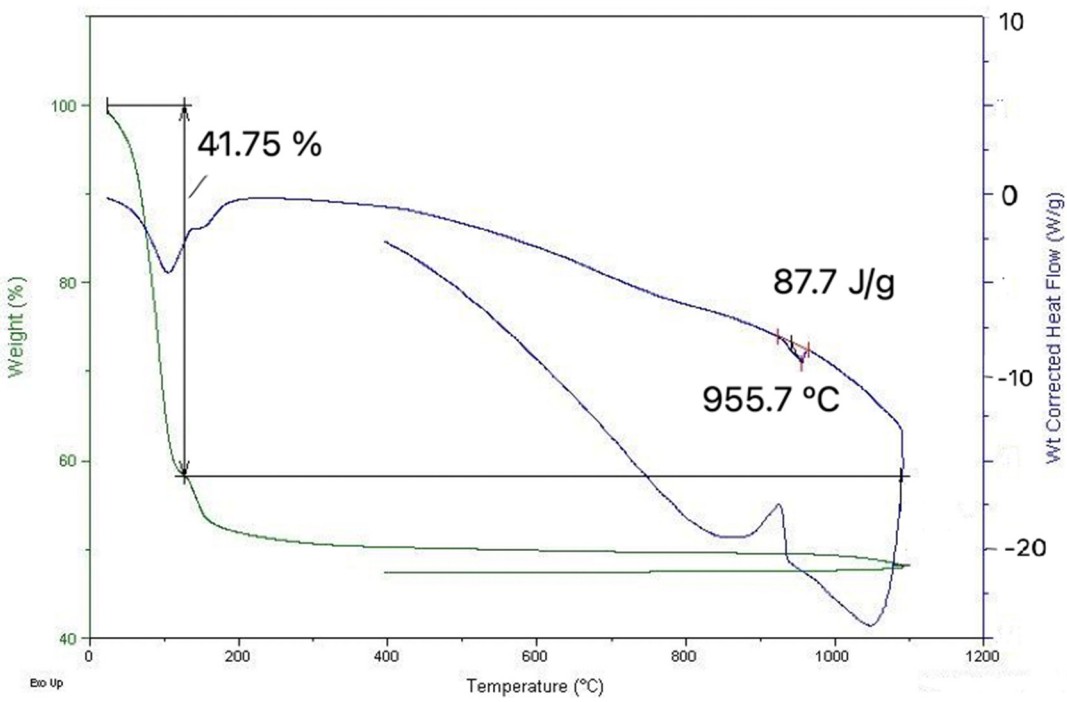

**Figure 9.** DTA/TGA graph.

This indicates that until around 130 °C, an important loss of weight occurs with an associated endothermic effect, which is due to moisture evaporation. Further weight loss occurs upon heating, representing around 10% of the initial mass of the sample. This can be explained by calcium sulfate and carbonate decompositions, the first one occurring at 150 °C, the other at much higher temperatures, being more pronounced after 700 °C. A reversible thermal effect is evident at around 950 °C, with an endothermic effect on heating. This suggests the melting of the material (solidification on cooling), which is coherent with the observation in electron microscopy.

### 3.6. Infrared Spectroscopy

Infrared spectra showed practically the same peaks and relative intensities. Figure 10 presents one of the spectra obtained for one sample dried at 105 °C (sample collected on 18.02.2022).

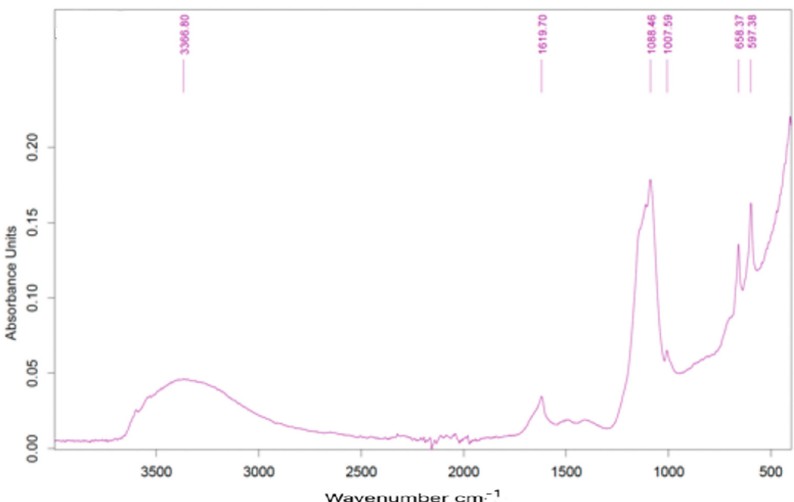

**Figure 10.** FTIR spectrum of dried sample.

Relevant peaks are present at 1619, 1068, 1007, 658 e 598 cm$^{-1}$. The band around 3400 cm$^{-1}$ is normally identified as evidence of water molecules, which may be due to the hydration of calcium sulfate. The peak at 1068 cm$^{-1}$ can be due to the presence of sulfates, the other peaks being unassigned to any type of specific compound.

The same sample calcined at 1000 °C presents the spectrum indicated in Figure 11.

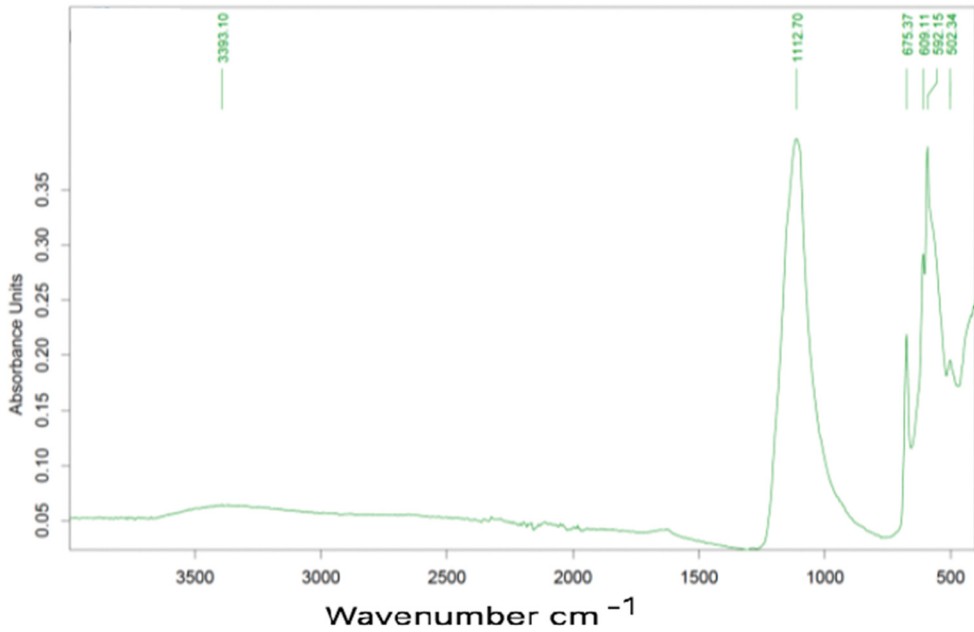

**Figure 11.** FTIR Spectrum of calcined sample.

Characteristic peaks are located at 1118, 658, 610, 592 e 502 cm$^{-1}$, all these correspond to gypsum.

*3.7. Water Leaching and Analysis of the Dissolved Solids*

Table 5 presents the total dissolved solids (TDS) obtained after water leaching when dried at 105 °C and calcined samples at 1000 °C, as well as their chemical composition. The eluates obtained for the dried samples presented a pH in the range of 6.3–6.5 while for the test with the calcined samples, the pH was in the range of 6.7–6.8, which, in both cases was somehow higher than the pH of the sludge, which was measured as 5.63.

**Table 5.** Results of the water leaching expressed in terms of total solids and their chemical composition.

| Sample | TDS (mg/kg) | S | Ca | Mg | K | Na | Mn | Ni | Mo |
|---|---|---|---|---|---|---|---|---|---|
| 14.02.2022 dried | 25,700 | 22.1 | 26.1 | 1.3 | 0.6 | 0.2 | 0.010 | 0.005 | 0.008 |
| 15.02.2022 dried | 25,600 | 21.2 | 21.8 | 2.4 | 1.2 | 0.4 | 0.26 | 0.032 | 0.006 |
| 16.02.2022 dried | 25,900 | 21.7 | 25.3 | 1.3 | 0.7 | 0.2 | 0.40 | 0.045 | 0.005 |
| 17.02.2022 dried | 26,200 | 22.0 | 24.6 | 1.5 | 1.0 | 0.2 | 0.50 | 0.071 | 0.004 |
| 18.02.2022 dried | 26,400 | 21.9 | 24.8 | 1.8 | 0.9 | 0.2 | 0.083 | 0.028 | 0.008 |
| Average dried | 25,960 | 21.8 | 24.5 | 1.7 | 0.9 | 0.2 | 0.251 | 0.036 | 0.006 |
| 14.02.2022 calc | 26,400 | 21.6 | 25.2 | < 0.1 | 1.8 | 2.0 | 0.002 | 0.001 | 0.034 |
| 15.02.2022 calc | 24,700 | 22.0 | 25.2 | < 0.1 | 2.2 | 2.8 | 0.001 | <0.001 | 0.054 |
| 16.02.2022 calc | 24,700 | 22.1 | 25.7 | < 0.1 | 1.8 | 1.8 | 0.001 | 0.001 | 0.090 |
| 17.02.2022 calc | 24,000 | 22.0 | 26.4 | < 0.1 | 1.4 | 1.5 | 0.002 | < 0.001 | 0.096 |

**Table 5.** *Cont.*

| Sample | TDS (mg/kg) | S | Ca | Mg | K | Na | Mn | Ni | Mo |
|---|---|---|---|---|---|---|---|---|---|
| 18.02.2022 calc | 23,900 | 22.2 | 26.7 | < 0.1 | 1.9 | 1.7 | 0.002 | 0.001 | 0.092 |
| Average calcined | 24,740 | 22.0 | 25.8 | < 0.1 | 1.8 | 2.0 | 0.002 | 0.001 | 0.073 |

Values are in weight % except for TDS.

These results indicate that around 2.5% of the material is leached, which is mainly composed of calcium sulfate. No Chromium, Iron or other metallic elements were detected in the dissolved solids. As the quantification limit for Iron and Chromium and the other metallic elements with higher atomic numbers than 19 is estimated as 0.001%, this means that these elements dissolved in less than 0.25 mg/kg. It appears that the calcining reduces the water leachability of Mg and Ni, while seems to increase the leachability for K, Na and Mo.

The expected releases in the eluates, based on the average results for the composition of the solid, and assuming that all sulfur is in the form of sulfates, may be calculated as presented in Table 6 in mg/kg of leached sample.

**Table 6.** Estimated concentration of the eluates obtained after water leaching.

| mg/kg | Dried | Calcined |
|---|---|---|
| Sulfates | 17,000 | 16,300 |
| Ca | 6400 | 6400 |
| Mg | 440 | <25 |
| K | 230 | 450 |
| Na | 50 | 490 |
| Mn | 65 | 0.5 |
| Ni | 9 | 0.25 |
| Mo | 1.6 | 18 |
| Cr | <0.25 | <0.25 |
| Fe | <0.25 | <0.25 |

*3.8. Ammonia Citrate Leaching Test*

Table 7 presents the results of the neutral ammonium citrate test, performed on the dried sludge and on the sample calcined at 1000 °C, this one after milling and sieving at less than 63 μm. The results for the elements are expressed in terms of mg for a kg of leached sample.

**Table 7.** Results of the ammonium citrate test.

| Parameter | Dried Sludge | Sludge Calcined at 1000 °C |
|---|---|---|
| Undissolved mass (%) | 52.8 | 52.7 |
| Final pH | 6.73 | 6.66 |
| Al (mg/kg) | 42 | 19 |
| Sb (mg/kg) | <2 | <2 |
| As (mg/kg) | <1 | 6 |
| Ba (mg/kg) | 4 | 4 |
| Be (mg/kg) | 0.02 | 0.02 |

**Table 7.** *Cont.*

| Parameter | Dried Sludge | Sludge Calcined at 1000 °C |
|---|---|---|
| B (mg/kg) | 7 | 6 |
| Cd (mg/kg) | <0.2 | <0.2 |
| Ca (mg/kg) | 55,400 | 55,600 |
| Cr (mg/kg) | 692 | 155 |
| Co (mg/kg) | 59 | <0.2 |
| Cu (mg/kg) | 779 | 10 |
| Fe (mg/kg) | 2840 | 56 |
| Pb (mg/kg) | 1.6 | 1.3 |
| Li (mg/kg) | <0.2 | <0.2 |
| Mg (mg/kg) | 530 | 81 |
| Mn (mg/kg) | 129 | 4 |
| Hg (mg/kg) | <1 | <1 |
| Mo (mg/kg) | 63 | 217 |
| Ni (mg/kg) | 2410 | 2 |
| P (mg/kg) | 34 | 443 |
| K (mg/kg) | 351 | 382 |
| Se (mg/kg) | <3 | <3 |
| Ag (mg/kg) | <0.5 | <0.5 |
| Na (mg/kg) | 242 | 471 |
| Th (mg/kg) | <1 | <1 |
| V (mg/kg) | 1 | 28 |
| Zn (mg/kg) | 24 | 13 |

It is clear that there is a strong degree of dissolution of Calcium. The analysis of the crystals obtained by evaporation of the eluate indicates that this element is in the form of calcium sulfate. In the test performed on the dried sludge, some heavy metals, such as Cr, Cu and Ni, are easily dissolved. This is particularly relevant when foreseeing the valorization of these types of sludges in agriculture, due to the detrimental effects of these toxic metals. However, in the calcined sludge, the degree of mobilization of these heavy metals is strongly reduced, while the release of molybdenum and phosphorus is clearly enhanced. This may be interesting in view of its possible use as a fertilizer component.

**4. Conclusions and Prospects for Valorization**

These characterization results indicate that the residue is a potential source of fluorine and calcium, and it also contains some interesting amounts of metals such as Nickel and Chromium. During a calcination procedure, there occurs a melting at around 950 °C, modifying relevantly some of the properties of the material.

In view of its possible use, we find that the presence of calcium fluoride is a major point of interest. As a matter of fact, this compound, also known in the industry as fluorspar, is a critical material in Europe. Some of the main uses are related to the making of hydrofluoric acid and in metallurgical operations, namely in the aluminum industry, as a flux compound for electrolysis, and in steelmaking, as a desulfurization agent [26]. Besides other minor uses, it can also be employed in glassmaking to produce opalescence [27].

For use in steelmaking, to modify the composition of ladle treatment slag to reduce viscosity and enhance the desulphurization efficiency [28], the high content of calcium sulfate in the sludge is detrimental. However, the removal of sulfur to produce more

concentrated calcium fluoride is possible by carbonation [29], for example, in sodium carbonate aqueous solutions. This route is presently being studied by our research team.

For use in glassmaking, however, the employment of the residue in its original form, just after drying, could be a possibility in relatively small amounts. The presence of calcium sulfate should not be a problem, as its presence in the chemical composition of glasses may be accepted in small amounts. This possibility justifies further research and the definition of which types of glasses could accept material with this composition.

Its use in agriculture is also a possibility. However, the release of toxic metals, especially Cr, Cu and Ni, would reduce the applicability of the residue in its original form. However, the strong decrease in the release of these metals when the material is previously calcined, and the fact that it releases some molybdenum, an interesting element for some cultures [30], makes this a possible route for valorization. In this case, the effect of pH on leaching and also the type of Cr ions that are released are relevant aspects to be taken into account.

The content of Nickel in the sludge presents a similar level as current ores [31]. In this view, the residue could be considered a feedstock for Nickel metallurgy. However, the limited quantity of these types of residues when compared to the size of the nickel extraction industry may be a weakness for this possible route.

A lower grade valorization of this residue could also be incorporated in concrete or in ceramics. In this case, special attention has to be paid to the leachability of heavy metals in the final product. As a matter of fact, the residues from this material are normally judged as inert. The incorporation of this residue should not modify this characteristic.

**Author Contributions:** Conceptualization, F.C. and N.M.G.P.; methodology, F.C.; investigation, F.C., P.B.T., N.C., J.F.G. and T.T.; writing, F.C. and P.B.T.; writing—review and editing, T.T.; supervision, F.C.; project administration, F.C.; funding acquisition, N.M.G.P. All authors have read and agreed to the published version of the manuscript.

**Funding:** This research was funded by Bollinghaus Steel, S.A.

**Institutional Review Board Statement:** Not applicable.

**Informed Consent Statement:** Not applicable.

**Data Availability Statement:** Not applicable.

**Acknowledgments:** The authors wish to acknowledge the participation of technicians Elsa Ribeiro, Leonor Carneiro and Miguel Abreu, from the University of Minho, in the running of some characterization activities.

**Conflicts of Interest:** The authors declare no conflict of interest. The funders had no role in the design of the study; in the collection, analyses, or interpretation of data; in the writing of the manuscript; or in the decision to publish the results.

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
