# Peer review of "Characterization of Stainless Steel Spent Pickling Sludge and Prospects for Its Valorization"

_metals, doi:10.3390/met12091539_

Round 1

Reviewer 1 Report

Dear Authors,

It is an excellent article that deserves to be published after corrections. I appreciated the presentation of the different results and methods in the manuscript, which is a help to the reader. Applications for the future are given at the end of the text.

Page 7. XRD. Indicate the compounds corresponding to the main peaks in Figure 3.
Page 8. XRD. Indicate the compounds corresponding to the main peaks in Figure 4.
Page 9. SEM.
Indicate the position of zones Z1 and Z2 in Figure 5.

Page 9. EDS. Indicate the elements corresponding to the peaks in Figure 6.

Page 10. SEM. Indicate the position of zones Z1, Z2, and Z3 in Figure 7.

Page 11. EDS. Indicate the elements corresponding to the peaks in Figure 8.

Page 12. DTA-TGA. The text cannot be read inside Figure 9.

Page 13. FTIR. Indicate the compounds corresponding to the main peaks in Figures 10, and 11.

Bibliographic references, ref. 21, the year is missing.

Author Response

Dear Reviewer,

We thank very much your opinion and suggestions. Regarding the mentioned points:

Page 7. XRD. Indicate the compounds corresponding to the main peaks in Figure 3.

Done
Page 8. XRD. Indicate the compounds corresponding to the main peaks in Figure 4.

Done
Page 9. SEM. Indicate the position of zones Z1 and Z2 in Figure 5.

Done

Page 9. EDS. Indicate the elements corresponding to the peaks in Figure 6.

Done

Page 10. SEM. Indicate the position of zones Z1, Z2, and Z3 in Figure 7.

Done

Page 11. EDS. Indicate the elements corresponding to the peaks in Figure 8.

done

Page 12. DTA-TGA. The text cannot be read inside Figure 9.

Just one of the peaks can be assigned (sulphates), the other may be linked to different types of bonds, not possible to identify solely by this technique. This has been mentioned in the text.

Page 13. FTIR. Indicate the compounds corresponding to the main peaks in Figures 10, and 11.

Bibliographic references, ref. 21, the year is missing.

Corrected

We included a new version for figure 2, because it was not easy to get copyright for the other figure. We did this new figure by our means, using data published in several works, listed in the reference 6. The purpose of this figure is just illustrative of the mechanism of water treatment.

We included also a new table (table 6), following a suggestion of other reviewer. Then the following table was numbered 7.

Reviewer 2 Report

General comments

The manuscript presents an interesting subject in the field of resource efficiency and addresses the critical raw material fluorspar. The lconsideration of the valorization potential of residual pickling sludge from steelmaking is an interesting issue in that respect.

Overall, the manuscript is well written. However, the experimental design could be partly better explained. A description of QA measures for the analyses is missing. The evaluation of the results and the conclusions could be improved.

The valorization potential is strongly dependent on the intended application and the possible release of hazardous substances in the application scenario. Unfortunately, the leaching tests haven’t been performed as required regarding the analysis of the eluates. Moreover, I would have expected an evaluation of the oxidation state of Cr since it has an important influence on the leaching behavior and toxicity of the eluates. As mentioned in the manuscript, the pH influence on the leaching behavior is crucial and therefore I would have expected more details on this aspect. Please add at least some information on the pH of the sludge and on the initial pH of the eluates. Only thorough evaluation of leaching data can identify the valorization potential in reuse scenarios such as in agriculture or construction. Please take into account that the leaching behavior may also change during an application period when the pH of the environment changes.

It would be helpful if the possible valorization options were not only listed in the conclusions but could be related to the experimental results of this study and compared to threshold values.

 Detailed comments

Detailed comments

Introduction

The introduction is informative and includes sufficient and current literature.

 Line 46

In order to don’t mix mining activities with the production of HF described in the previous sentence, it’s probably better to speak of exporter instead of producer here.

Line 57

Better put steel into singular.

Line 74

Do you mean solubility in water? CaSO4 is much better soluble than CaF.

Lines 76-83

Which pH is usually obtained by the neutralization process?

Mention the meaning of the dots in figure 2.

Line 104

Correct the typo: replace “in” by “is”

Materials and Methods

Line 126

Please describe the applied procedures for sampling in the plant and obtaining representative subsamples in the lab. A particle size distribution or at least the maximum particle size would be of interest already here (even if there is some information in 3.4).

Line 136

Correct “a oven” to “an oven”.

Line 154

Correct “Phases quantification were…” to “Phases quantification was…”.

Lines 184-188

There are a lot of deviations from EN 12457-4. Please justify! The standard procedure requires an end-over-end tumbler (5 rpm - 10 rpm) or roller-table inducing rotation of the bottle at about 10 rpm. A rotary jar at a speed of 0.5 rpm is definitely not equivalent. What type of filtration was performed? Why didn’t you measure the concentration of elements in the eluates using ICP-OES or similar? When 90 g of the sample were taken as required - evaporation of the eluate is a lot of effort and may be associated with losses of substances.

Lines 192-196

Please mention the sample intake and the type of analysis done for the leachates.

Results and Discussion

Table 2

Please mention for the convenience of the readers that the results were obtained using XRF in the legend of the table. Standard deviations should be added to support the statement on homogeneity.

Figure 3

The main peaks of the phases could be marked by letters or numbers.

Table 3

A content of 17% of Fe2O3 is indicated in table 2. However, there is only one Fe-bearing phase determined by XRF with minor content. Do you have an explanation? Amorphous Fe? Table 4 shows about 20% of iron oxides in the calcined samples which fits better to the total content when you subtract oxygen uptake by the calcination.

Figure 5

The indicators are hardly to spot (similar for Figure 7). Please use another font size and color.

An example of a tubular-like crystal in an image of higher magnification may reveal the crystal shape and provide another proof for the assumption of CaSO4 as the phase of these crystals.

Figure 6 and 8

The font size of the indicators is too small. Think about enlarging the spectra.

Line 291

Correct “analysis” to “analyses”

Line 343

Can you possibly observe a proof for the oxidation of iron?

Line 344

Modify “observed” to “observation” to improve the grammar.

Lines 369/370

The header of 3.7 and the text should be revised since the eluates were actually not analysed but the evaporated samples. Can you additionally calculate back the concentration in the eluates and add them to the table as that is the usual result of a leaching test which enables the evaluation of the environmental behavior by comparison with limit values? The limit of quantification would have been much better if the eluates were analyzed using ICP-OES or ICP-MS.

Table 5

Why wasn’t Fe measured? What was the detection limit for Cr.

Lines 380/381

Can you attribute the effects to phases and their solubility?

Line 387

Please explain in which condition the eluates were analyzed.

Lines 395/398

These statements are not supported by data. Cr and Cu are not listed in table 5 (might be retained in the filter cake?).

Conclusions

Line 402

Correct “indicates” to “indicate”.

Line 441

Correct: “these materials” or “this material”

Author Response

Dear reviewer,

We thank you very much your comments and suggestions for revision. As for the points you mentioned, we present the following modifications and comments:

The valorization potential is strongly dependent on the intended application and the possible release of hazardous substances in the application scenario. Unfortunately, the leaching tests haven’t been performed as required regarding the analysis of the eluates. Moreover, I would have expected an evaluation of the oxidation state of Cr since it has an important influence on the leaching behavior and toxicity of the eluates.

In the water leaching, soluble Cr was not detected, as mentioned. So, it was impossible to identify the oxidation state. We mentioned, however, the quantification limits to try to demonstrate this aspect.

As mentioned in the manuscript, the pH influence on the leaching behavior is crucial and therefore I would have expected more details on this aspect. Please add at least some information on the pH of the sludge and on the initial pH of the eluates.

We agree with this. So, we included the value for the pH of the sludge and the pH of the eluates.

Only thorough evaluation of leaching data can identify the valorization potential in reuse scenarios such as in agriculture or construction. Please take into account that the leaching behavior may also change during an application period when the pH of the environment changes.

We agree with your opinion. So, a comment has been included: “In this case, effect of pH on leaching and also the type of Cr ions that are released are relevant aspects to be taken into account”

It would be helpful if the possible valorization options were not only listed in the conclusions but could be related to the experimental results of this study and compared to threshold values.

This could be a different way of organizing the paper. However, we followed the structure of the papers as asked by the editor, and we found easier to inlude them in the conclusions part.

 Detailed comments

Detailed comments

Introduction

The introduction is informative and includes sufficient and current literature.

 Line 46

In order to don’t mix mining activities with the production of HF described in the previous sentence, it’s probably better to speak of exporter instead of producer here.

We modified to be clearly expressed that the production is of fluorspar, to not be confusing with HF

Line 57

Better put steel into singular.

done

Line 74

Do you mean solubility in water? CaSO4 is much better soluble than CaF.

 We change the sentence to

In both cases precipitating compounds, calcium fluoride and calcium sulphate, as effectively they are not both equally insoluble.

Lines 76-83

Which pH is usually obtained by the neutralization process?

Between 8 – 8,5. Indicated in the text.

Mention the meaning of the dots in figure 2.

Figure 2 has been changed by one made by ourselves, because we could not find the way of getting copyright for the other figure. There are no dots now.

Line 104

Correct the typo: replace “in” by “is”

 done

Materials and Methods

Line 126

Please describe the applied procedures for sampling in the plant and obtaining representative subsamples in the lab.

The way for the sampling in the plant and in the lab has been now described.

A particle size distribution or at least the maximum particle size would be of interest already here (even if there is some information in 3.4).

We didn’t the particle size distribution measurement. However, as can be seen by the SEM image, there are large tubular like particles, reaching sometimes more than 50 micrometers, with very fine round shaped ones, sometimes in the range of 1 micrometer. We indicated that in the text (3.4).

Line 136

Correct “a oven” to “an oven”.

 done

Line 154

Correct “Phases quantification were…” to “Phases quantification was…”.

done

Lines 184-188

There are a lot of deviations from EN 12457-4. Please justify! The standard procedure requires an end-over-end tumbler (5 rpm - 10 rpm) or roller-table inducing rotation of the bottle at about 10 rpm. A rotary jar at a speed of 0.5 rpm is definitely not equivalent. What type of filtration was performed? Why didn’t you measure the concentration of elements in the eluates using ICP-OES or similar? When 90 g of the sample were taken as required - evaporation of the eluate is a lot of effort and may be associated with losses of substances.

We confirmed with the laboratory that performed the leaching test, and effectively the speed was not 0,5 but 7,5. The filtration was made with 0,45 mm PTFE membranes under vacuum.

Lines 192-196

Please mention the sample intake and the type of analysis done for the leachates.

 Done

Results and Discussion

Table 2

Please mention for the convenience of the readers that the results were obtained using XRF in the legend of the table. Standard deviations should be added to support the statement on homogeneity.

 done

Figure 3

The main peaks of the phases could be marked by letters or numbers.

 done

Table 3

A content of 17% of Fe2O3 is indicated in table 2. However, there is only one Fe-bearing phase determined by XRF with minor content. Do you have an explanation? Amorphous Fe? Table 4 shows about 20% of iron oxides in the calcined samples which fits better to the total content when you subtract oxygen uptake by the calcination.

 Yes, we thing this suggest the presence of amorphous Fe phases. We indicated that in the text: “The fact that no relevant peaks were identified for iron bearing phases suggests that this element is present, in the sludge, in amorphous form”

Figure 5

The indicators are hardly to spot (similar for Figure 7). Please use another font size and color.

done

An example of a tubular-like crystal in an image of higher magnification may reveal the crystal shape and provide another proof for the assumption of CaSO4 as the phase of these crystals.

Unfortunately we didn’t get a higher magnification image, as the material is not enough conductive to obtain a good quality picture at higher magnification..

Figure 6 and 8

The font size of the indicators is too small. Think about enlarging the spectra.

 Corrected the fonts.

Line 291

Correct “analysis” to “analyses”

 corrected

Line 343

Can you possibly observe a proof for the oxidation of iron?

 No, we didn’t tried to prove the degree of oxidation of iron. In principle it is Fe3+, but we did not determined.

Line 344

Modify “observed” to “observation” to improve the grammar.

 done

Lines 369/370

The header of 3.7 and the text should be revised since the eluates were actually not analysed but the evaporated samples. Can you additionally calculate back the concentration in the eluates and add them to the table as that is the usual result of a leaching test which enables the evaluation of the environmental behavior by comparison with limit values?

Header corrected. We included a new table (now table 6) with the calculation of the expected concentration. Table 6 was modified for 7.

The limit of quantification would have been much better if the eluates were analyzed using ICP-OES or ICP-MS.

We agree. But, as we couldn’t know a priori which were the elements to be analysed and as the XRF spectrometer allows a very low quantification limit (except for Mg), our option was to do like this. The other parameters, especially those related to environmental legislation should be all at very low levels, probably not relevant for the purposes of the work.

Table 5

Why wasn’t Fe measured? What was the detection limit for Cr.

Detection limit for Cr and Fe: 0,001 % in the XRF spectrometer.

Lines 380/381

Can you attribute the effects to phases and their solubility?

No. We find it would be speculative with the data we have.

Line 387

Please explain in which condition the eluates were analyzed.

It is explained that the results are for the dissolved solids, not for the eluates. With the new table, according to your suggestion, we find that this is more clear now.

Lines 395/398

These statements are not supported by data. Cr and Cu are not listed in table 5 (might be retained in the filter cake?).

This means that Cr and Cu do not dissolve in water. While, in the citrate test, much more aggressive, they dissolve.

Conclusions

Line 402

Correct “indicates” to “indicate”.

 done

Line 441

Correct: “these materials” or “this material”

Corrected

We included a new version for figure 2, because it was not easy to get copyright for the other figure. We did this new figure by our means, using data published in several works, listed in the reference. The purpose of this figure is just illustrative of the mechanism of water treatment.

Round 2

Reviewer 2 Report

I still have some concerns with the experimental approach of the measurement of the eluate composition. The sensitivity of XRF measurements is not as good as for ICP-OES or ICP MS. However, to correct this, new experiments were needed.

In Table 6 no concentrations are provided but releases. Concentrations in the eluate are given in e.g. mg/l, releases which refer to the sample intake in mg/kg.

line 369: correct measure to measured

Author Response

Dear reviewer,

Thanks again for your comments. 

We have changed the word "concentrations" for "releases", effectively more adequate, and made the correction in line 369.

Concerning the concern about the measurement of the eluate composition, our objective was not to measure all the potential impacts in terms of environment, but mainly to evaluate the degree of solubility of the residue, which is relevant for the definition of valorization routes. But, of coarse, we follow your concern, and agree that a measurement of the eluate composition by other, more usual, methods (ICP and others), would be more adequate. If you find this is determinant for the publication, we can yet do this measurement and improve the paper. but this will take around 30 days to obtain the results. Best regards.
